# Effects of Left Ventricular Unloading on Cardiac Function, Heart Failure Markers, and Autophagy in Rat Hearts with Acute Myocardial Infarction

**DOI:** 10.3390/ijms26094422

**Published:** 2025-05-06

**Authors:** Ryota Azuma, Yasushige Shingu, Jingwen Gao, Satoru Wakasa

**Affiliations:** Department of Cardiovascular Surgery, Faculty of Medicine and Graduate School of Medicine, Hokkaido University, Kita 15 Nishi 7, Kitaku, Sapporo 060-8638, Japan; r_azuma_0612@yahoo.co.jp (R.A.); gaojingwen0404@outlook.com (J.G.); wakasa@med.hokudai.ac.jp (S.W.)

**Keywords:** LV unloading, acute myocardial infarction, autophagy

## Abstract

Percutaneous ventricular assist devices are utilized in cases of cardiogenic shock following acute myocardial infarction (AMI). However, the mechanism underlying the beneficial effects of LV unloading in AMI remains unclear. This study aimed to examine the impact of LV unloading on cardiac function, heart failure markers, and protein degradation (autophagy and ubiquitin–proteasome system: UPS) post AMI in rats. Nine-week-old male Lewis rats were randomized into non-AMI, AMI, non-AMI with LV unloading, and AMI with LV unloading groups. LV unloading was achieved through heterotopic heart–lung transplantation. Rats were euthanized 2 and 14 days after the procedure. Cardiac functional assessment was performed using Langendorff heart perfusion. RT-PCR and Western blot analyses were conducted using the LV myocardium. The rate pressure product was comparable between the non-AMI with LV unloading group and the AMI with LV unloading at 14 days. The atrial natriuretic factor tended to be suppressed by LV unloading. LV unloading had reducing effects on the expressions of p62, selectively degraded during autophagy, both 2 and 14 days after AMI. There was no effect on the parameters for the UPS. LV unloading has a mitigating effect on the deterioration of cardiac function following AMI. Autophagy, which was suppressed by AMI, was ameliorated by LV unloading.

## 1. Introduction

In the United States, acute myocardial infarction (AMI) resulted in over 100,000 deaths in 2020 [1]. LV dysfunction following AMI precipitates cardiogenic shock, with a reported incidence rate of 4.4% among AMI cases [2]. Cardiogenic shock is associated with a high mortality rate ranging from 40% to 50% [3]. Recently, mechanical cardiac support devices have been increasingly used for cardiogenic shock to augment stroke volume and/or unload the LV. The appropriate type or combination of support devices can be selected with the guidance of right heart catheterization [4]. However, the relative risk of mortality has not been reduced even by the use of mechanical circulatory support devices [3]. Although the use of mechanical circulatory support in cardiogenic shock is recommended as Class II a in the ESC guidelines, the level of evidence is low (level C) [5].

The effectiveness of utilizing Impella^®^ (Abiomed Inc., Danvers, MA, USA), a percutaneous ventricular assist device, remains unclear [6]. The DanGer trial is the only study that has recently shown that Impella^®^ reduces mortality after AMI-related cardiogenic shock [7]. This device pumps blood from the LV to the ascending aorta, thereby reducing the preload of the LV, a process known as “LV unloading”. This process contributes to the reduction in myocardial mechanical work, decreases myocardial oxygen demand, and is expected to facilitate the recovery of cardiac function [8]. Using a sheep model, Meyns B. et al. reported that full Impella^®^ support during 60 min of ischemia and 120 min of reperfusion reduced myocardial oxygen consumption and decreased infarct size by 75% [8]. Similarly, in dog experiments conducted by Saku K. et al., total Impella^®^ support initiated 60 min after the onset of ischemia and continued through 60 min of reperfusion reduced infarct size by 80%, with cardiac function being well preserved [9]. However, these studies only examined the acute effects of Impella^®^ in reperfusion models. The effects of prolonged LV unloading in cases of more severe infarction or no-reflow remain unknown.

Elucidating the mechanism of LV unloading effects in AMI is necessary for decision-making regarding the use of this device. AMI induces cardiac remodeling [10], whereas LV unloading reduces cardiomyocyte size [11]. Counteracting cardiac remodeling depends on protein degradation, which occurs via two major systems: autophagy and the ubiquitin–proteasome system (UPS). Autophagy represents a critical metabolic process responsible for breaking down senescent or damaged proteins and organelles into amino acids and fatty acids, facilitating energy production and recycling [12]. It has been established that basal autophagy is essential for maintaining normal cardiac function [13]. The UPS is a large, ATP-dependent cascade system that operates in a stepwise manner. The initial stage in the UPS degradation of proteins involves labeling through the covalent attachment of ubiquitin. Subsequently, the tagged protein undergoes degradation by the 26S proteasome [14]. We considered that autophagy or the UPS plays a key role in the effects of prolonged LV unloading for AMI.

The objective of this study was to examine the impact of LV unloading after AMI on cardiac function, heart failure markers, autophagy, and the UPS. Although several studies have used complete LV unloading models in animals, which were unable to accurately replicate real clinical situations [15,16], this study employed a partial LV unloading model through heterotopic heart and lung transplantation in rats to better mimic the clinical situation.

## 2. Results

### 2.1. Partial LV Unloading Model Effectively Reduced LV Size

The baseline echocardiographic data were comparable among all groups (Appendix A). This partial LV unloading model reduced LV sizes both in the acute and subacute phases. None of the parameters showed the interaction between AMI and LV unloading (Appendix A). Functional assessment through echocardiography under varying LV loading conditions is not ideal; therefore, we utilized Langendorff heart perfusion in the subsequent section.

### 2.2. LV Unloading Tended to Attenuate Cardiac Functional Deterioration After AMI

Figure 1 presents the cardiac functional parameters of Langendorff heart perfusion. None of the parameters showed interaction between AMI and LV unloading in the 2-day models. For the maximum dPdt and rate pressure product (RPP), the interactions between AMI and LV unloading were not statistically significant; however, the effect sizes were medium in the 14-day model (*p* = 0.18, partial η^2^ = 0.067; *p* = 0.14, partial η^2^ = 0.085, respectively). Furthermore, in the post hoc analyses, dPdt and RPP values were not significantly lower in the AMI with LV unloading group than the non-AMI with LV unloading group in the 14-day model. These data may suggest an attenuating effect of LV unloading on cardiac functional deterioration on post-AMI day 14.

### 2.3. LV Unloading Did Not Affect Myocardial Remodeling After AMI

Figure 2 indicates the LV weight and histological data. AMI increased LV weight and myocyte area in both the 2-day and 14-day models. Conversely, LV unloading reduced these parameters only in the 14-day model. For the LV weight and myocyte, the interactions between AMI and LV unloading were not statistically significant and the effect sizes were small in the 14-day model (*p* = 0.18, partial η^2^ = 0.0092; *p* = 0.12, partial η^2^ = 0.018, respectively). Fibrosis was observed in the LV unloading groups in the 14-day model. Fibrosis did not exhibit an interaction between AMI and LV unloading in either the 2-day or 14-day models. These data suggest that LV unloading did not affect the impact of AMI on myocardial hypertrophy and fibrosis.

### 2.4. LV Unloading Tended to Attenuate the Increased Fetal Gene Expression After AMI

The expression levels of the following genes were evaluated using LV myocardium (Figure 3): α-myosin heavy chain (MHC), βMHC, atrial natriuretic factor (ANF), brain natriuretic peptide (BNP), and sarco/endoplasmic reticulum Ca^2+^-ATPase (SERCA2). For the ANF, the interaction between AMI and LV unloading was not statistically significant; however, the effect size was medium in the 14-day model (*p* = 0.080, partial η^2^ = 0.10). In the post hoc analysis, a significant difference in ANF levels was found only between the non-AMI group and the AMI group in the 14-day model. These data suggest that increased expression of ANF, a fetal gene, was suppressed 14 days after AMI in the presence of LV unloading. MHC and BNP did not show an interaction between AMI and LV unloading in the 2- and 14-day models, while BNP levels increased following AMI. For the SERCA2, the interaction between AMI and LV unloading was not statistically significant; however, the effect size was large in the 2-day model (*p* = 0.071, partial η^2^ = 0.17).

### 2.5. LV Unloading Attenuated Expressions of an Autophagy-Related Protein, p62 After AMI

The expression of microtubule-associated light chain (LC) 3-II, which corresponds to the amount of autophagosomes, can either increase or decrease following the activation of autophagy. In contrast, p62 is selectively degraded during autophagy and decreases when autophagy is activated.

Figure 4 indicates the expressions of autophagy-related proteins. In LC3-II expression, neither main effects nor interactions were observed. In contrast, significant interactions in p62 between AMI and LV unloading were observed and the effect sizes were large in both 2-day and 14-day models (*p* = 0.001, partial η^2^ = 0.25; *p* = 0.039, partial η^2^ = 0.20, respectively). The levels of p62 were higher in the unloading groups compared to the non-unloading groups in both the 2-day and 14-day models. The p62 levels were significantly higher in the AMI group than in the non-AMI group in the 14-day model. Thus, these data suggest that LV unloading significantly impacted the effects of AMI on autophagy activation, while LV unloading itself decreased autophagy flux.

Phosphorylated AMP-activated protein kinase (AMPK)/total AMPK, an energy sensor, was also significantly affected by LV unloading, and showed a significant interaction between AMI and LV unloading in the 2-day model. This ratio was significantly higher in the AMI group than in the non-AMI group in the 14-day model. These data suggest that LV unloading significantly impacted the effects of AMI on energy preservation.

### 2.6. LV Unloading Did Not Affect the Parameters of the UPS and Protein Synthesis

Figure 5 presents the parameters of the UPS and protein synthesis. To assess ubiquitin-ligating (E3) enzymes, which play key roles in recognizing specific protein substrates in the UPS, the gene expressions of F-box protein 32 (*Atrogin1*) and Muscle RING-finger protein-1 (*MuRF1*) were examined. LV unloading and AMI influenced pAKT/AKT and *MuRF1* levels, respectively, in the 14-day model. The following protein or gene expressions showed no interaction between AMI and LV unloading in the 2- and 14-day models: K48, a marker of polyubiquitination; Atrogin1 and MuRF1; and the ratio of pAKT/AKT, a marker of protein synthesis. These data suggest that LV unloading did not impact the effects of AMI on the UPS and protein synthesis.

## 3. Discussion

We have demonstrated that LV unloading tended to attenuate cardiac functional deterioration post AMI. Furthermore, LV unloading impacted the effects of AMI on myocardial autophagy activation. The attenuation of cardiac functional deterioration by LV unloading was observed only at the later stage of AMI (14 days), suggesting that the effects of LV unloading after complete coronary occlusion without reperfusion require time to manifest.

### 3.1. Autophagy in AMI and LV Unloading

Activation of autophagy improves cardiac function after AMI; autophagy is upregulated and adaptive during ischemia. Kanamori et al. reported that cardiomyocyte autophagy was upregulated after starvation, resulting in decreased infarct size during AMI for up to 24 h [17]. Moreover, Sciarretta et al. found that chronic ischemia suppressed autophagy flux and that oral administration of trehalose, an autophagy activator, reduced cardiac dysfunction and remodeling through autophagy activation after 4 weeks of AMI [18].

The change in autophagy by LV unloading remains a topic of controversy. In investigations of autophagy in human samples of LV unloading using LVADs for non-ischemic etiology, Martin et al. showed activation of autophagy, while Kassiotis et al. reported downregulation of autophagy by LV unloading [19,20]. In a rat model of complete LV unloading [21], Brinks et al. demonstrated no changes in autophagy markers after LV unloading for up to 60 days [22]. Conversely, Cao et al. reported an increase in LC3-II levels after 4, 7, and 14 days of complete LV unloading in mice [16]. However, it remains unclear whether autophagy flux was activated in this unloading model for normal mice.

In our model, autophagy was attenuated by AMI (elevated p62 on day 14, which supports the previous study [18], and this attenuation was abolished in the presence of LV unloading. Furthermore, dPdt and RPP did not decrease on day 14 compared to those in non-AMI UL. We speculate that preventing the attenuation of autophagy after AMI may have contributed to preserving cardiac function through LV unloading. On the other hand, LV unloading itself was associated with reduced autophagy flux, as indicated by increased p62 levels. Therefore, further studies are needed to investigate whether enhancing autophagy pharmacologically could provide additional benefits for AMI in combination with LV unloading. If autophagy activity can be properly regulated in clinical settings, it may promote earlier recovery of cardiac function and potentially shorten the duration of LV assist device use after AMI.

### 3.2. Cardiac Remodeling and Ubiquitin Ligases

Regarding LV unloading, several reports have investigated myocyte remodeling and ubiquitin ligases, *Atrogin1* and *MuRF1*, also known as atrophy-related genes. Baskin et al. reported that complete LV unloading induces cardiac atrophy after seven days [15]. In their study, cardiac hypertrophy was observed in *Atrogin1* knock-down mice, leading to the conclusion that *Atrogin1* is essential for atrophy following LV unloading. Diakos et al. demonstrated that in human LV cardiomyopathy, gene expressions of *Atrogin1* and *MuRF1* remained unchanged following LVAD support [23].

In our study, LV unloading did not affect *Atrogin1* and *MuRF1* either. We attribute this discrepancy from previous studies to differences in species, procedures (complete or partial unloading), and the duration of unloading. LV unloading induces cardiac remodeling, yet the relationship to ubiquitin ligases remains unclear.

### 3.3. Cardiac Fibrosis After LV Unloading

Several studies have reported that cardiac fibrosis occurs following LV unloading. Annette et al. demonstrated that in patients with LVADs, cardiac fibrosis increased within six months after device implantation [24]. Similarly, Schaefer et al. reported that in a rat heterotopic heart transplantation model, suppression of the biomechanical stress regulators FHL1 and FHL2 was associated with myocardial fibrosis [25]. However, the exact mechanism is still not fully understood.

In our study, LV unloading also led to cardiac fibrosis, consistent with previous findings. This result may be associated with cardiac disfunction in the LV unloading groups. However, we did not investigate the associated genes, proteins, or molecular pathways involved. Further studies are required to elucidate the mechanisms underlying cardiac fibrosis induced by LV unloading.

### 3.4. Limitations

A potential limitation in our study is the influence of cardiac “denervation” resulting from heart transplantation, which may affect the observed effects of LV unloading. Previous experimental studies have reported that cardiac denervation can lead to cardiac atrophy and dysfunction [26,27]. However, in human heart transplantation, the heart does not exhibit atrophy [28]. Thus, the effects of cardiac denervation might be minimal in this model. Another limitation is the inability to assess longitudinal changes in cardiac function in individual animals, as Langendorff parameters were used as the primary endpoints. Lastly, although heart failure symptoms or survival could support the biochemical findings, they do not reflect the function of the unloaded hearts in recipient rats. Therefore, we did not assess them.

### 3.5. Conclusions

In a rat partial LV unloading model, there was an indication of the attenuating effects of LV unloading on cardiac functional deterioration post AMI. Autophagy flux in the myocardium was diminished by AMI, and this attenuation was counteracted by LV unloading. Further studies are required to elucidate how modulation of autophagy by LV unloading following AMI contributes to the preservation of cardiac function.

## 4. Materials and Methods

### 4.1. Experimental Design

Figure 6 shows the experimental protocol. Nine-week-old male Lewis rats (Japan SLC, Inc., Hamamatsu, Japan) were randomized into the following four groups (*n* = 4–6 per group): non-AMI, AMI, non-AMI with LV unloading, and AMI with LV unloading. The operation was conducted under general anesthesia via a single intramuscular injection of ketamine (90 mg/kg; Ketalar; Daiichi Sankyo Pharmaceutical, Tokyo, Japan) and xylazine (10 mg/kg; Selactar; Bayer Yakuhin, Osaka, Japan). Rats were euthanized 2 (acute: Figure 6A) and 14 (subacute: Figure 6B) days after the operation. In this model, the 14-day time point was selected based on a previous study of partial LV unloading [29]. Echocardiography was conducted before the operation (baseline) and before the euthanization. Ex vivo cardiac functional assessment was performed using Langendorff perfusion. Histological examination, RT-PCR, and Western blot were carried out using the LV myocardium excised after Langendorff perfusion.

### 4.2. AMI and Heterotopic Heart and Lung Transplantation

All procedures were performed under general anesthesia with intubation and mechanical ventilation. AMI was created by ligating the left anterior descending artery (LAD) using 7-0 polypropylene (Ethicon) after left lateral thoracotomy. In the non-AMI group, only thoracotomy was performed.

LV unloading was performed using the heterotopic heart–lung transplantation procedure, as previously described by Ibrahim et al. [30]. This model was defined as partial unloading of the LV. Briefly, donor rats received an injection of heparin (1000 U; Mochida Pharmaceutical Co., Ltd., Tokyo, Japan) via the inferior vena cava. The donor heart was administrated cold 50 mL St. Thomas II solution to induce cardiac arrest and excised with the lungs. The ascending aorta of the donor heart was then anastomosed to the abdominal aorta of the recipient rat (Figure 7).

The time course of the procedure in the AMI with LV unloading group was as follows: ligation of the LAD in the donor rats, preparation of the abdominal vessels in the recipient rats, injection of cardioplegia via the descending aorta of the donors, and subsequent heart–lung transplantation. The entire procedure was completed within 2 h, with the ischemic time of the donor hearts kept within 35 min. 

### 4.3. Echocardiography

Under anesthesia, transthoracic or transabdominal echocardiography was performed using a Sonos 5500 ultrasound system equipped with a 12 MHz phased-array transducer (Philips Medical Systems, Andover, MA, USA). The heart rate (HR), LV end-diastolic dimension (LVEDD), and LV end-systolic dimension (LVESD) were measured using M-mode tracings obtained from the long-axis view of the LV. Fractional shortening was calculated using the LVEDD and LVESD.

### 4.4. Langendorff Heart Perfusion

Secobarbital sodium (150 mg/kg: Aional; Nichi-Iko Pharmaceutical Co., Ltd., Toyama, Japan) was intraperitoneally injected for euthanasia. The heart was promptly resected and was perfused using the Langendorff system with a Krebs–Henseleit buffer containing the following substances (Sigma-Aldrich, Darmstadt, Germany): NaCl 128 mM, KCL 5.0 mM, MgSO_4_ 1.3 mM, KH_2_PO_4_ 1.0 mM, CaCl_2_ 2.5 mM, NaHCO_3_ 15 mM, and glucose 5.0 mM. The buffer was oxygenated with 95% O_2_ and 5% CO_2_. The perfusion pressure was set at 60 mmHg.

A balloon was inserted into the LV via the left atrium to acquire pressure data, which were recorded using PowerLab (ADInstruments, Dunedin, New Zealand) and analyzed with Lab-Chart (ADInstruments). The LV end-diastolic pressure was set at 5–10 mmHg. At the end of the 30 min perfusion, the following parameters were assessed: HR, coronary flow, LV developed pressure (LVDP), maximal rates of increase and decrease in velocity of LV pressure (maximum and minimum dPdt), and rate pressure product (RPP = HR × LVDP).

### 4.5. Histological Examination

The LV myocardium at the level of the papillary muscle was fixed in 3.5% neutral formalin, embedded in paraffin, and sectioned at 5-μm intervals. The cardiomyocyte area was measured using hematoxylin–eosin-stained sections. The cross-sectional area of cardiomyocyte was measured by selecting 100 randomly chosen oval-shaped cardiomyocytes with a nucleus in the non-infarcted area using ImageJ software 1.54f (NIH). The percentage of fibrotic area in the non-infarcted area was calculated using Masson’s trichrome staining to assess fibrosis. Ten frames were randomly selected from each slice and analyzed using ImageJ.

### 4.6. RT-PCR

Total RNA was isolated from frozen myocardial tissue samples by the High Pure RNA Tissue Kit (Roche, Basel, Switzerland). cDNA was transcribed from the total RNA using the Transcriptor First Strand cDNA Synthesis Kit (Roche). Quantitative real-time RT-PCR was performed using the FastStart Essential DNA Probes Master (Roche) and RealTime ready assay (Roche). PCR amplification was conducted at a volume of 20 μL using LightCycler Nano (Roche) according to the manufacturer’s instructions. The results were normalized to the transcription of S29, which exhibited comparable expression levels among groups.

### 4.7. Western Blotting

To evaluate myocardial autophagy, we assessed the protein levels of LC3-II and p62 [31]. Autophagy consists of four main stages (autophagic flux): (1) initiation, (2) autophagosome formation, (3) fusion of autophagosomes with lysosomes, and (4) breakdown and recycling of autophagic bodies. LC3-II is derived from LC3-I (its cytosolic form) and binds to autophagosomal membranes, reflecting the quantity of autophagosomes. However, LC3-II may be elevated by inhibiting its breakdown stage. Conversely, p62, selectively taken up by autophagosomes and degraded during autophagic body breakdown, serves as an indicator of autophagic degradation; reduced p62 expression indicates increased autophagic flux.

Electrophoresis and blotting were performed by a semidry Western blot apparatus (Mini-PROTEAN Tetra Cell; Bio-Rad, Hercules, CA, USA). Electrophoresis was undertaken using sodium dodecyl sulfate polyacrylamide gel (12% Mini-PROTEAN TGX; Bio-Rad). The proteins were blotted onto a polyvinylidene difluoride membrane (Immobilon; Millipore, Burlington, MA, USA) and incubated with primary antibodies (LC3B and p62: Abcam; AMPK, pAMPK, K48, protein kinase B (AKT), pAKT, and GAPDH: Cell Signaling Technology (CST, Danvers, MA, USA) and secondary antibodies (anti-rabbit IgG; CST). The bands were detected through chemiluminescence (ECL Prime; cytiva, Marlborough, MA, USA) and semi-quantified using JustTLC (Sweday, Södra Sandby, Sweden). The band intensity was normalized by GAPDH.

### 4.8. Statistical Analysis

All data are reported as mean and SEM. Baseline echocardiographic data were analyzed using one-way ANOVA to detect differences. Two-way ANOVA was employed for all parameters except baseline. When acknowledging the interaction (AMI × unloading), it was considered that LV unloading has an impact on the course of AMI. A post hoc Bonferroni test was conducted because of only a small number of comparisons. Statistical significance was determined at *p* < 0.05 using Prism 9.5.1 (GraphPad, San Diego, CA, USA). Effect sizes (partial η^2^) were calculated using Python 3.11.12 (Python Software Foundation, Wilmington, DE, USA) in Google Colaboratory (Google Research, Mountain View, CA, USA, https://colab.research.google.com/, accessed on 30 April 2025) for several important parameters.

## Figures and Tables

**Figure 1 ijms-26-04422-f001:**
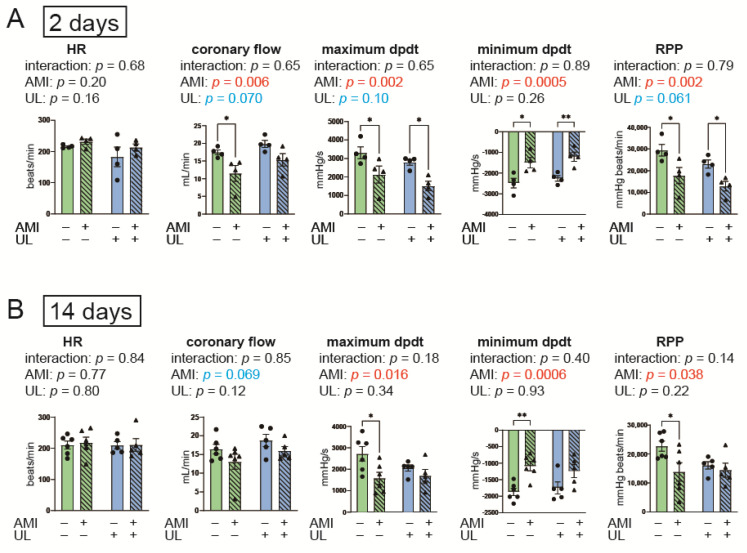
Cardiac functional parameters of Langendorff heart perfusion in the 2- (**A**) and 14-day (**B**) models. AMI, acute myocardium infarction; HR, heart rate; RPP, rate pressure product; UL, unloading. *n* = 4–6 for each group. Data are expressed as mean and SEM. * *p* < 0.05 and ** *p* < 0.01 (post hoc Bonferroni). The *p* values in the figure are those of the interaction (AMI × UL) and main effects (AMI or UL) in the two-way ANOVA.

**Figure 2 ijms-26-04422-f002:**
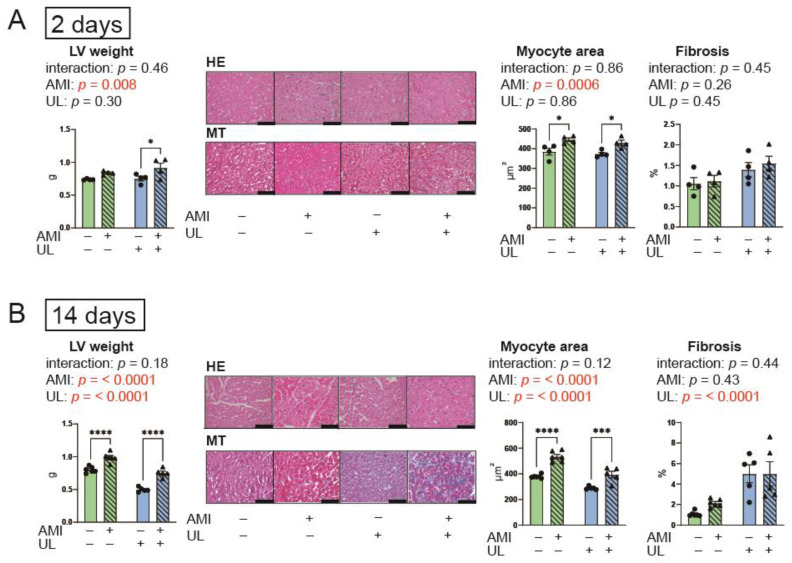
LV weight and histological data in the 2- (**A**) and 14-day (**B**) models. AMI, acute myocardium infarction; HE, hematoxylin eosin; MT, Masson’s trichrome; UL, unloading. *n* = 4–6 for each group. Data are expressed as mean and SEM. * *p* < 0.05, *** *p* < 0.001, and **** *p* < 0.0001 (post hoc Bonferroni). The scale bars indicate 100 μm. The *p* values in the figure are those of the interaction (AMI × UL) and main effects (AMI or UL) in the two-way ANOVA.

**Figure 3 ijms-26-04422-f003:**
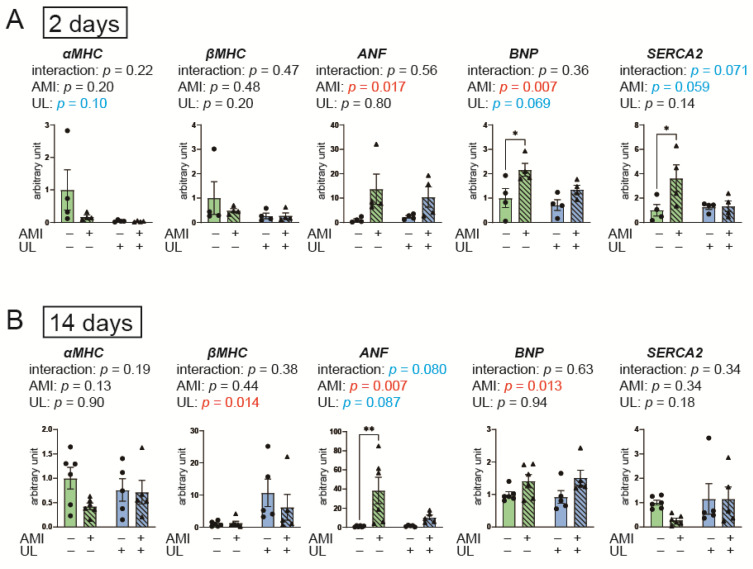
Myocardial gene expressions of MHC, ANF, BNP, and SERCA2 in the 2- (**A**) and 14-day (**B**) models. AMI, acute myocardium infarction; ANF, atrial natriuretic factor; BNP, brain natriuretic peptide, MHC, myosin heavy chain; SERCA2, sarco/endoplasmic reticulum Ca2+-ATPase; UL, unloading. *n* = 4–6 for each group. Data are expressed as mean and SEM. * *p* < 0.05 and ** *p* < 0.01 (post hoc Bonferroni). The *p* values in the figure are those of the interaction (AMI × UL) and main effects (AMI or UL) in the two-way ANOVA.

**Figure 4 ijms-26-04422-f004:**
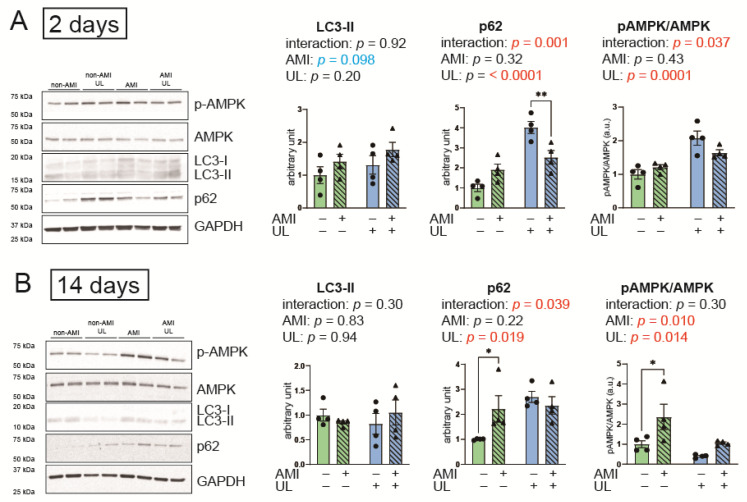
Myocardial expressions of autophagy-related proteins by Western blot in the 2- (**A**) and 14-day (**B**) models. AMI, acute myocardium infarction; AMPK, AMP-activated protein kinase; GAPDH, glyceraldehyde-3-phosphate dehydrogenase; LC3-II, microtubule-associated light chain 3 II; pAMPK, phosphorylated AMPK; UL, unloading. *n* = 4 for each group. Data are expressed as mean and SEM. * *p* < 0.05 and ** *p* < 0.01 (post hoc Bonferroni). The *p* values in the figure are those of the interaction (AMI × UL) and main effects (AMI or UL) in the two-way ANOVA.

**Figure 5 ijms-26-04422-f005:**
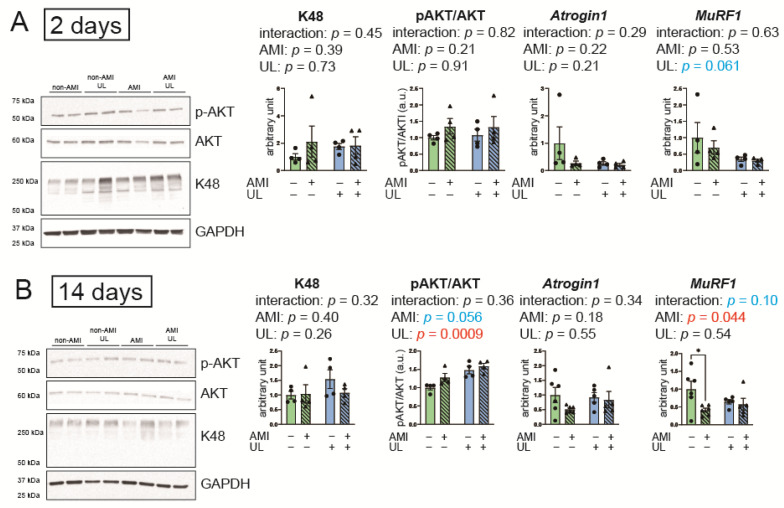
Parameters of UPS and protein synthesis in the 2- (**A**) and 14-day (**B**) models. K48 is a marker of polyubiquitination. The ratio of pAKT/AKT is a marker of protein synthesis. *Atrogin1* and *MuRF1* are ubiquitin ligases. K48 and AKT were assessed with Western blot; Atrogin1 and MuRF1 were evaluated with RT-PCR. AMI, acute myocardium infarction; GAPDH, glyceraldehyde-3-phosphate dehydrogenase; UL, unloading. *n* = 4–6 for each group. Data are expressed as mean and SEM. * *p* < 0.05 (post hoc Bonferroni). The *p* values in the figure are those of the interaction (AMI × UL) and main effects (AMI or UL) in the two-way ANOVA.

**Figure 6 ijms-26-04422-f006:**
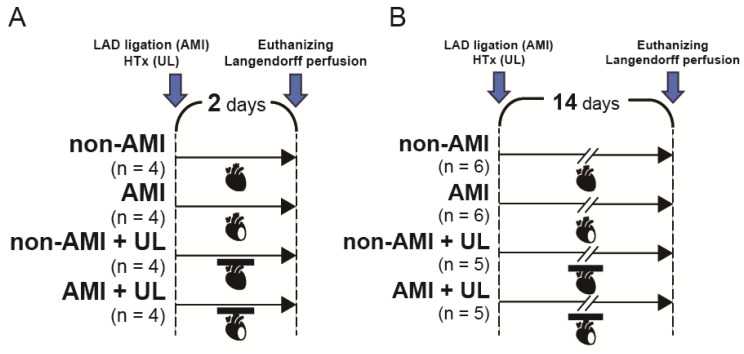
Experimental protocol of the 2- (**A**) and 14-day (**B**) models. Nine-week-old male Lewis rats were randomized into four groups (*n* = 4–6 per group): non-AMI, AMI, non-AMI with LV unloading, and AMI with LV unloading. Rats were euthanized 2 and 14 days after the operation and ex vivo cardiac functional assessment was performed by Langendorff system. AMI, acute myocardium infarction; HTx, heart transplantation; LAD, left anterior descending artery; UL, unloading.

**Figure 7 ijms-26-04422-f007:**
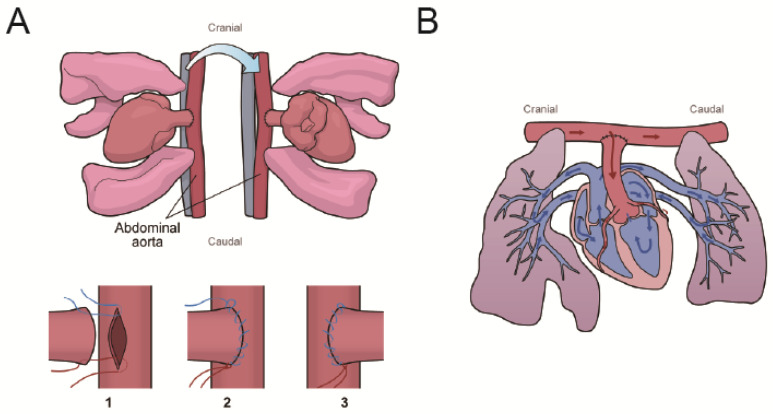
Schema of the partial LV unloading procedure (**A**) and blood flow in the donor heart (**B**). (**A**) The donor heart was extracted with lungs. Two sutures for the abdominal and ascending aorta were ligated (7-0 polypropylene) at both ends of the incision (1). The left side of the anastomosis was created by a running suture (2). The donor heart and lungs were turned over to the left side. The right side of the anastomosis was created by a running suture (3). (**B**) The LV was partially unloaded because the LV was loaded only by the coronary perfusion volume. The red and blue arrows indicate oxygenated and unoxygenated blood, respectively.

## Data Availability

The original contributions presented in this study are included in the article/Appendix A. Further inquiries can be directed at the corresponding author.

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
