# Peer review of "Effects of Left Ventricular Unloading on Cardiac Function, Heart Failure Markers, and Autophagy in Rat Hearts with Acute Myocardial Infarction"

_ijms, 2025, doi:10.3390/ijms26094422_

Round 1

Reviewer 1 Report

Comments and Suggestions for Authors

Azuma et al. have used heterotrophic heart-lung transplantation model to study effects of partial unloading after experimental AMI in rats.

The rate-pressure product was comparable between the non-AMI with LV unloading group and the AMI with LV unloading at 14 days. Atrial natriuretic factor tended to be suppressed by LV unloading. LV unloading had reducing effects of the expressions of p62, selectively degraded during autophagy, both 2 and 14 days after AMI. There was no effect on the parameters for the UPS.

Authors conclude that LV unloading has a mitigating effect on the deterioration of cardiac function following AMI. Autophagy which was suppressed by AMI was ameliorated by LV unloading.

This model is suitable for studying pronged effects of unloading after myocardial infarction. Manuscript is clearly written, and results are well presented.

Major comments:

  1. This paper focuses on basic mechanisms of cardiac remodeling after myocardial infarction. There might be limited direct relevance to clinical assist devices. I suggest shortening review of the clinical trials in introduction.
  2. Results Fig 2 show significant increases in the level of fibrosis after unloading. The mechanism of the increase is unclear. This phenomenon should be discussed.
  3. Results Fig2 show increase in SERCA2 after 2 days, but a decrease after 14 days (green bars). Unloading, seems to prevent changes. Please, check if the statistical analysis is performed adequately.

Author Response

We sincerely thank the Reviewer for the helpful comments to improve our manuscript.

Comments:

  1. This paper focuses on basic mechanisms of cardiac remodeling after myocardial infarction. There might be limited direct relevance to clinical assist devices. I suggest shortening review of the clinical trials in introduction.

Author’s response:

Thank you for your valuable suggestion. However, in this paper, we prefer not to reduce the statements of clinical trials in the Introduction section. This is because the motivation for our study is grounded in clinical experience, and Reviewer 2 suggested expanding the information regarding clinical presentations that require left ventricular (LV) unloading.

  1. Results of Fig 2 show significant increases in the level of fibrosis after unloading. The mechanism of the increase is unclear. This phenomenon should be discussed.

Author’s response:

As the reviewer pointed out, the mechanism underlying the increase in fibrosis remains unclear. In LVAD patients, cardiac fibrosis has been reported to increase by six months after LVAD implantation (Annette H. et al., 2006). In a rat heterotopic heart transplantation model, suppression of biomechanical stress regulators FHL1 and FHL2 was correlated with myocardial fibrosis (Schaefer A. et al., 2019). However, the exact mechanism is still not fully understood. We have added this discussion to the Discussion section (lines 257 - 268).

  1. Results of Fig3 show increase in SERCA2 after 2 days, but a decrease after 14 days (green bars). Unloading, seems to prevent changes. Please, check if the statistical analysis is performed adequately.

Author’s response:

We have re-checked the statistical analysis regarding the expression of SERCA2, and we confirm that it was performed correctly. As the reviewer pointed out, unloading appears to prevent changes in SERCA2 expression at both 2 and 14 days. However, the effect of unloading seems to be more pronounced at 2 days (interaction P = 0.071, partial η² = 0.17, large effect) than at 14 days (interaction P = 0.34, partial η² = 0.04, small effect).

Reviewer 2 Report

Comments and Suggestions for Authors

In this basic science study Dr. Ryota Azuma, and colleagues aimed to examine the impact of left ventricular (LV) unloading with a partial LV unloading model (through heterotopic heart and lung transplantation) on cardiac function, heart failure markers, and protein degradation (autophagy and ubiquitin-proteasome system: UPS) post-acute myocardial infarction (AMI) in rats.

The authors showed that LV unloading has a mitigating effect on the deterioration of cardiac function following AMI. Autophagy, suppressed by AMI, was ameliorated by LV unloading.

The study research topic warrants careful consideration, due to strong interest in cellular biology during LV unloading in post-AMI clinical course.

Overall, this is a quite nicely written article.

However, it has some limitations that should be addressed from the authors:

  • The major limitation of the study is the very small sample size, especially for a study using two-way ANOVA, which reduces the power to detect interaction effects and increases the risk of both Type I and Type II errors. Confidence intervals are not reported, making it difficult to assess the precision of the estimates.

  • In the Introduction section, it could be useful to provide any information about the clinical presentations requiring the LV unloading. This section should be expanded, with a mention about the right heart catheterization use in order to choose the best hemodynamic strategy in patients with RV-PGD. In this respect, the authors should cite one of the most recent papers: (Manzi, L et al, Diagnostics (Basel, Switzerland), 14(2), 136. https://doi.org/10.3390/diagnostics14020136).

  • Many outcomes (functional, molecular, histological) are tested across multiple groups and time points, yet there is no correction beyond Bonferroni for multiple endpoints (e.g., no FDR or Holm-Bonferroni adjustment). This raises concern for false positive results due to multiple testing.

  • Several results (e.g., ANF expression, SERCA2 levels) are described as “borderline significant” without meeting conventional p < 0.05 thresholds, yet they are discussed as trends.

  • Functional data are collected ex vivo (Langendorff perfusion) rather than by serial in vivo echocardiography at multiple time points. This limits the ability to track dynamic changes in individual animals.

  • Including survival or functional outcome endpoints (e.g., heart failure symptoms, exercise capacity) could be helpful to complement biochemical findings.

  • How the authors suggest that these clinical results could impact on future studies or on current clinical practice? Please, expand this section and further discuss how the findings in a rat heterotopic model may translate to human AMI patients supported by devices like Impella.

Author Response

We sincerely thank the Reviewer for the helpful comments to improve our manuscript.

  1. The major limitation of the study is the very small sample size, especially for a study using two-way ANOVA, which reduces the power to detect interaction effects and increases the risk of both Type I and Type II errors. Confidence intervals are not reported, making it difficult to assess the precision of the estimates.

Author’s response:

We appreciate the reviewer’s comment on this point. The magnitude of the effect is often more informative than the precision of the estimate, especially when assessing biological or experimental relevance. Since our statistical analysis was mainly based on ANOVA, reporting partial η² as an effect size is a standard and widely accepted approach in this context. Moreover, effect sizes are less sensitive to sample size and provide a clear indication of the strength of differences between experimental conditions, which we believe is particularly useful for interpreting our findings. Accordingly, we have added the effect sizes (partial η²) in the text, instead of reporting confidence intervals, and we revised the methods section of statistical analysis (lines 411 - 412).

  1. In the Introduction section, it could be useful to provide any information about the clinical presentations requiring the LV unloading. This section should be expanded, with a mention about the right heart catheterization use in order to choose the best hemodynamic strategy in patients with RV-PGD. In this respect, the authors should cite one of the most recent papers: (Manzi, L et al, Diagnostics (Basel, Switzerland), 14(2), 136. https://doi.org/10.3390/diagnostics14020136).

Author’s response:

We appreciate the reviewers’ valuable advice. In accordance with the recommendation, we have added comments on the use of right heart catheterization (RHC) and the timing of LV unloading, based on the most recent literature, to the Introduction section (lines 34 - 37).

  1. Many outcomes (functional, molecular, histological) are tested across multiple groups and time points, yet there is no correction beyond Bonferroni for multiple endpoints (e.g., no FDR or Holm-Bonferroni adjustment). This raises concern for false positive results due to multiple testing.

Author’s response:

Thank you for your valuable comment. We agree that the Holm–Bonferroni procedure generally provides greater statistical power than the traditional Bonferroni method and is considered a more powerful alternative for controlling the family-wise error rate (FWER).

However, in our study, the number of post hoc comparisons was strictly limited to pairwise comparisons within each main factor (i.e., two comparisons per factor). Given this small number of comparisons, the difference in power between the Bonferroni and Holm–Bonferroni methods becomes negligible.

As noted by Abdi (2007), “when only a small number of comparisons are made, the Bonferroni correction is generally adequate and is often preferred for its simplicity and transparency, despite being more conservative than necessary.” In line with this, we chose the Bonferroni correction to maintain interpretability and because its conservativeness does not significantly compromise power in our context. Nonetheless, we appreciate the reviewer’s suggestion and will clarify this rationale in the revised manuscript to improve transparency (line 410).

  1. Several results (e.g., ANF expression, SERCA2 levels) are described as “borderline significant” without meeting conventional p < 0.05 thresholds, yet they are discussed as trends.

Author’s response:

To avoid potential misunderstanding, we changed the phrase from “borderline significant” to “not statistically significant.” However, to emphasize the trends, we have added the effect sizes to the sentence (lines 134 - 136, 141 - 143).

  1. Functional data are collected ex vivo (Langendorff perfusion) rather than by serial in vivo echocardiography at multiple time points. This limits the ability to track dynamic changes in individual animals.

Author’s response:

We are also interested in the dynamic changes of unloading in individual animals. However, we consider that cardiac function in this model can only be evaluated using Langendorff perfusion, as the loading conditions are not consistent in vivo. We have added this limitation regarding the inability to assess dynamic changes in cardiac function to the manuscript (lines 276 - 278).

  1. Including survival or functional outcome endpoints (e.g., heart failure symptoms, exercise capacity) could be helpful to complement biochemical findings.

Author’s response:

We fully agree with the reviewer’s comment. However, in this study, the symptoms observed in recipient rats do not reflect the functional outcome of the unloaded hearts. We have added this limitation to the manuscript (lines 278 - 280).

  1. How the authors suggest that these clinical results could impact on future studies or on current clinical practice?

Author’s response:

In this study, LV unloading after AMI impacts autophagy activity and contributed to the suppression of cardiac dysfunction. However, LV unloading itself was associated with reduced autophagy flux, as indicated by increased p62 levels. Therefore, further studies are needed to investigate whether enhancing autophagy pharmacologically could provide additional benefits in combination with LV unloading. If autophagy activity can be properly regulated in clinical settings, it may promote earlier recovery of cardiac function and potentially shorten the duration of LV assist device use after AMI. We have added this point to the Discussion section (lines 237 - 243).